# Pervious Pavement Blocks Made from Recycled Polyethylene Terephthalate (PET): Fabrication and Engineering Properties

**Byung-Hyun Ryu [1], Sojeong Lee [2] and Ilhan Chang [3],*** 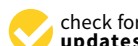

[1]   Department of Future Technology and Convergence Research, Korea Institute of Civil Engineering and Building Technology, Goyang-si 10233, Korea; tnt306@kict.re.kr
[2]   School of Engineering and Information Technology, University of New South Wales (UNSW), Canberra 2612, Australia; sojeong.lee@student.unsw.edu.au
[3]   Department of Civil System Engineering, Ajou University, Suwon-si 16499, Korea
*   Correspondence: ilhanchang@ajou.ac.kr; Tel.: +82-31-219-1534

**Abstract:** The importance of permeable and pervious pavements in reducing urban stormwater runoff and improving water quality is growing. Here, a new pervious pavement block material based on recycled polyethylene terephthalate (PET) waste is introduced, which could contribute to reducing global plastic waste via PET's utilization for construction material fabrication. The engineering properties and durability of recycled PET aggregate (RPA) pervious blocks are verified through flexural tests, in situ permeability tests, clogging tests, and freeze-thaw durability tests, and their cost-effectiveness is assessed by comparison with existing permeable/pervious pavers. Their engineering and economic characteristics confirm that the RPA pervious blocks are suitable for use in urban paving.

**Keywords:** pervious pavement block; recycled PET aggregate (RPA) block; flexural strength; permeability; freezing and thawing

## 1. Introduction

Permeable pavement, including pervious concrete, porous asphalt, and permeable interlocking pavers, has been widely studied for the management of urban surface runoff and stream erosion problems [1–3], groundwater recharge [4], and prevention of pollutant inflow into water systems in urban water quality control [5]. The increasing trend in urban flooding and geotechnical engineering hazards due to climate change [6] motivates research towards the development of environmentally friendly permeable pavement materials such as recycled concrete aggregates [2,7], seashells [8,9], and furnace residues (e.g., fly ash and bottom ash) [10,11].

The rapid global growth of plastic production (288 million metric tons (Mt) in 2012; a 620% increase since 1975) is accompanied by an increasing plastic waste and pollution problem, with 35% of plastic waste being from the food packaging and beverage industries [12]. This increase in plastic waste causes serious environmental threats, especially to oceans and marine ecosystems [13,14]. The Convention on Biological Diversity report of 2012 stated that all sea turtles, 45% of marine mammal species, and 21% of seabirds are at risk because of oceanic plastic pollution [15]. Ingested and inhaled microplastics are also a threat to human health [16]. Despite the significant threats posed by plastic waste and pollution, the global total of such waste is estimated to increase to 12,000 Mt by 2050, with 4977 t accumulated in 2015 alone [17]. Attempts have therefore been made to utilize plastic wastes for engineering construction and building purposes [18–21]. Previous studies of plastic waste recycling have involved laboratory tests of engineering properties and in situ performance. In this study, synthetic resin

waste was used as the main binder in the production of pervious blocks from aggregates of recycled polyethylene terephthalate (PET) waste. The engineering performance of the blocks (flexural strength, in situ permeability, clogging, and freeze–thaw durability) was tested experimentally.

## 2. Materials and Methods

### 2.1. Materials

PET is the most common thermoplastic resin used for food and beverage packaging, including plastic bottles [22], and thus PET chips were used to produce pervious blocks in this study. PET has the chemical composition $(C_{10}H_8O_4)_n$, a melting temperature of 250 °C, and a density of 1.35 g cm$^{-1}$ [23]. Used PET bottles were collected from a waste recycling plant in Incheon, Republic of Korea, and shredded into small chips of equivalent radius of 10 ± 5 mm to minimize microparticles (Figure 1a). The resulting PET chips were washed with water and dried for mixing. Coarse-grained aggregates having grain sizes of 2 × 6 mm, specific gravity ($G_s$) of 2.69, moist adsorption capacity of 0.51%, and unit weight of 1.46 g cm$^{-3}$ were used to produce recycled PET aggregate (RPA) blocks.

### 2.2. Fabrication of RPA Blocks

RPA blocks were fabricated by preheating aggregate in a heating muller mixer at up to 270 °C (Figure 1b). Clean PET chips were added with a 7.5% PET/aggregate mass ratio and mixed for 20 s at 30 rpm at 270 °C ± 5 °C. Preliminary mixing and molding (compacting) trials for different PET contents (2.5%, 5.0%, 7.5%, and 10.0%) indicated the 7.5% PET/aggregate mass ratio, which resulted in a 30% PET/aggregate volume ratio, to be the most feasible RPA block mixing condition in terms of thorough mixing and highest final density (2.13 g cm$^{-3}$) after molding. During heated mixing (Figure 1c), the PET chips melted and coated the surfaces of the aggregate particles. After mixing, the melted PET/aggregate mixture was poured quickly into a cuboid mold (200 mm long × 200 mm wide × 80 mm deep), compressed with a 40 MPa overburden pressure, and cooled to room temperature (Figure 1d). The block fabrication process (Figure 1a–d) was controlled to provide a void ratio of 0.2–0.3. The average RPA block density was maintained at 2.13 g cm$^{-3}$.

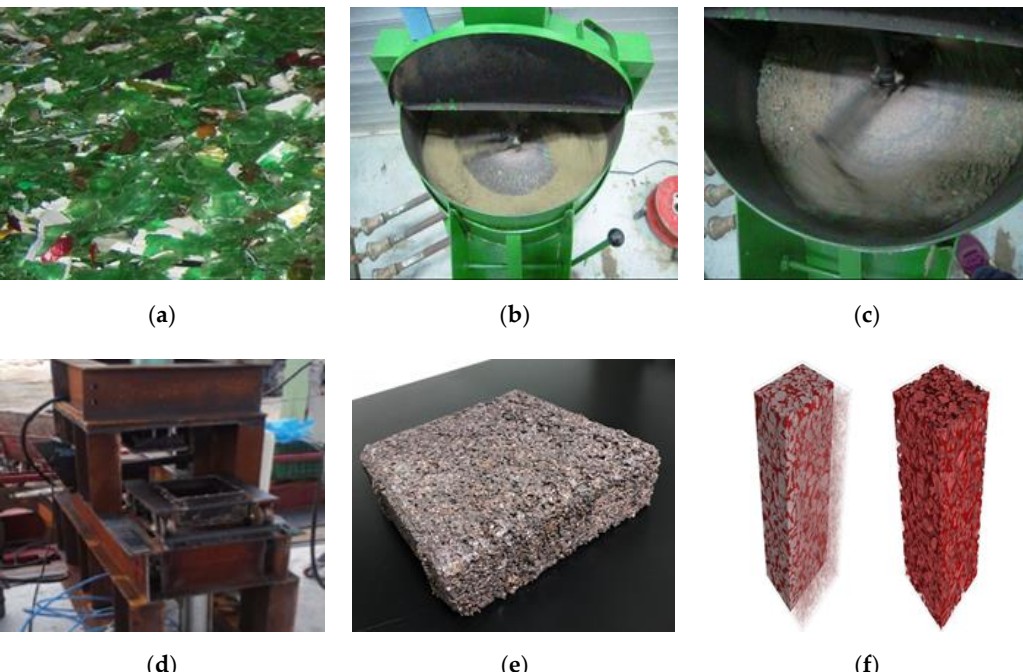

(a)　　　　　　　　　　　(b)　　　　　　　　　　　(c)

(d)　　　　　　　　　　　(e)　　　　　　　　　　　(f)

**Figure 1.** Overall procedure for recycled polyethylene terephthalate (PET) aggregate (RPA) block fabrication: (**a**) chopped PET chips; (**b**) aggregate heating; (**c**) RPA chips heated aggregate mixing; (**d**) shape molding with pressure; (**e**) fabricated RPA block; (**f**) X-ray CT profile of RPA block specimens.



## 2.3. Material Testing

### 2.3.1. Flexural Strength Test

The flexural strength test involved the application of a transverse load through a steel bearing plate (6 mm thick) along with the center line of an RPA block supported by two rods 175 mm apart, as per American Society for Testing and Materials (ASTM) C67 requirements [24]. The modulus of rupture at the plane of failure (*S*) was calculated by Equation (1):

$$S \ (\mathrm{Pa}) = \frac{3Wl}{2bd^2} \tag{1}$$

where $W$ = maximum load (N), $l$ = support spacing (175 mm), $b$ = net width (200 mm), and $d$ = depth (80 mm). The test was repeated with five samples to ensure consistency.

### 2.3.2. In Situ Permeability Test

Surrounding disturbed or unstable soil may cause clogging of pavement surfaces, reducing permeability. In situ infiltration testing was undertaken using a single-ring infiltrometer to infiltrate 3.6 kg of water after prewetting the test block, following ASTM C1701 requirements [25]. The penetration time was measured at 0.1 s intervals for 2 min, and the infiltration rate (*I*) was calculated using Equation (2):

$$I \left(\mathrm{mm \ h^{-1}}\right) = \frac{K \times M}{D^2 \times t} \tag{2}$$

where $M$ = mass of the infiltrated water (3.6 kg), $D$ = inner diameter of the infiltration ring (300 mm), $t$ = time (s) required for of the total water ($M$) to infiltrate, and $K$ = conversion coefficient (4,583,666,000 $\mathrm{mm^3 \cdot s \ (kg \cdot h)^{-1}}$). The test was repeated with three different samples to obtain a reliable average value.

### 2.3.3. Sand-Clogged Permeability Test

Laboratory-based sand-clogged permeability tests were performed under controlled clogging conditions that were based on an earlier study [26]. The RPA blocks were placed in a leveled supporting mold on a shaking table. Sand clogging of intergranular pore spaces in RPA blocks was achieved through a two-stage procedure: (1) 24 g of 'jumumjin' sand ($C_u$ = 1.94, $C_c$ = 1.09, $e_{max}$ = 0.89, $e_{min}$ = 0.64, $G_s$ = 2.65, $D_{50}$ = 0.52 mm, USCS = SP) was spread evenly over a dry RPA block and vibrated at 60 Hz for 30 s to allow the sand particles to penetrate the block; (2) water (400 mL) was poured onto the block, followed by a further 30 s of vibration, and permeability was assessed by the time required for the water to penetrate through the block. The test was conducted with three different samples to obtain a reliable average value.

### 2.3.4. Freeze–Thaw Durability Test

Cubed RPA specimens were used in rapid freeze–thaw cycles in accordance with ASTM C666 [27]. Hot-mixed PET/aggregate was poured into acrylic 80-mm-cubed molds, compressed, and cooled for hardening. Ten such RPA samples were prepared and placed in a freezing chamber with water, with a thermometer attached to the center of the top of each block. Freeze–thaw cycles were applied by freezing at −18 °C and thawing at +4 °C. The freezing stage was applied for 3 h and the thawing stage for 1 h, as per ASTM C666. A total of 120 freeze–thaw cycles were performed over 20 days. Durability was assessed by measuring the mass loss of each block after each of the 30 freeze–thaw cycles by removing loose particles and spalled materials via washing. Washed particles and spalled materials were collected and dried in an oven to determine the mass loss [27]. All ten samples were subjected to 120 freeze–thaw cycles.

## 3. Engineering Properties of RPA Blocks

The engineering properties (flexural strength, in situ permeability, sand-clogged permeability, and freeze–thaw durability) of the RPA block assessed via standard test methods [24,25,27] are summarized in Table 1 and compared to general design criteria for permeable pavements [28–32]. Details of each engineering property are described in the following sections.

**Table 1.** Engineering properties of the RPA block assessed by this study and relevant international design criteria.

| Properties | RPA Block | | International Criteria | |
| --- | --- | --- | --- | --- |
| | Test Results | Test Method | Recommendation | Institution |
| Flexural strength | 5.2 MPa | ASTM C67 | 1.0–3.8 MPa | Portland Cement Association (USA) |
| In situ permeability | 2365.2 mm/h | ASTM C1701 | 828.0 mm/h [2] | New Jersey Stormwater (USA) |
| | | | 360.0 mm/h [2] | Seoul Metropolitan Government (Korea) |
| Sand-clogged permeability | 2.93 to 1.63 mm/s | ASTM C1701 [1] | 0.14 mm/s [3] | Interlocking Concrete Pavement Institute (USA) |
| Durability (freezing and thawing) | 2.0% mass loss [4] per 120 cycles on average | ASTM C666 | 1% or less per 50 cycles | ASTM C936 |

[1] Sand-clogged permeability test was conducted according to ASTM C1701. [2,3] Minimum requirements for permeable concrete and porous asphalt. [4] Average mass loss after 120 repeated freeze–thaw cycles.

### 3.1. Flexural Strength

Flexural strength is one of the most important criteria in the evaluation of structural properties of pavements. All five samples had *S* values of >5.0 MPa, averaging 5.2 MPa, thus satisfying the flexural strength criterion of 1.0–3.8 MPa for pervious concrete [28]. Furthermore, the flexural strength of the RPA blocks compared well with that of ordinary concrete pavement (4.5 MPa), pervious concrete blocks (1.7–3.8 MPa), and permeable pavement based on recycled glass (3.4 MPa), as summarized in Figure 2 [33–35].

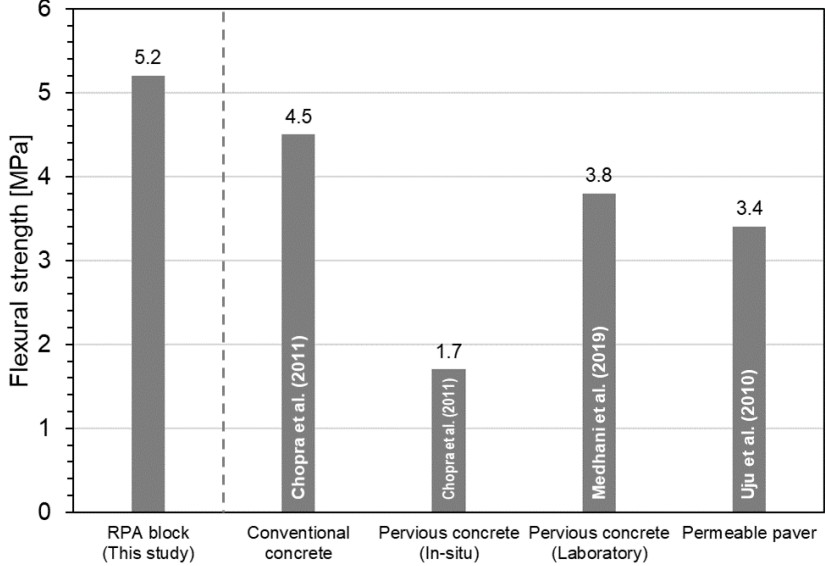

**Figure 2.** Flexural strength of the RPA block and comparison with other pervious pavement materials.

### 3.2. In Situ Permeability and Sand-Clogged Permeability

The in situ permeability of RPA blocks averaged 0.657 mm s$^{-1}$, satisfying minimum design criteria (Table 1) for pervious blocks (Korea, 0.1 mm s$^{-1}$; USA, 0.23 mm s$^{-1}$) [29,30] and exceeding that of previously produced permeable pavement materials (Figure 3) [36,37]. This level of ground infiltration capability indicates that RPA blocks should be effective for urban stormwater control.

The nonclogged laboratory-based permeability (2.93 mm s$^{-1}$) was higher than the in situ permeability because of the absence of an underlying soil layer, although this decreased to 1.63 mm s$^{-1}$ with sand clogging. The sand-clogged permeability satisfies the minimum infiltration rate of 0.14 mm s$^{-1}$ recommended by the US Interlocking Concrete Pavement Institute [31]. The sand-clogged permeability of the RPA blocks was also improved when compared to values reported in previous studies (Figure 4) [1,38,39], with lower maintenance (e.g., cleaning or vacuuming) requirements and cost as previously anticipated.

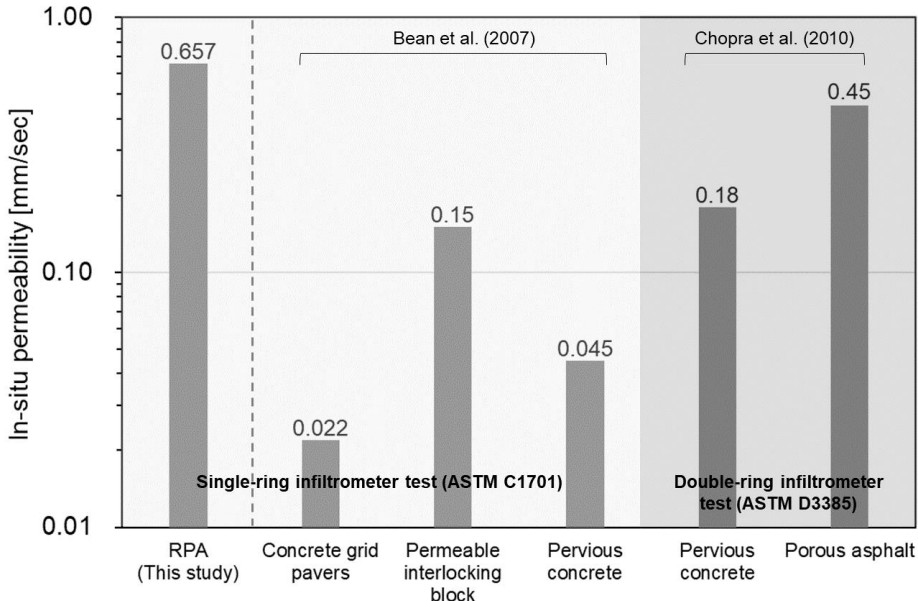

**Figure 3.** In situ permeability of the RPA block and comparison with other pervious pavement materials.

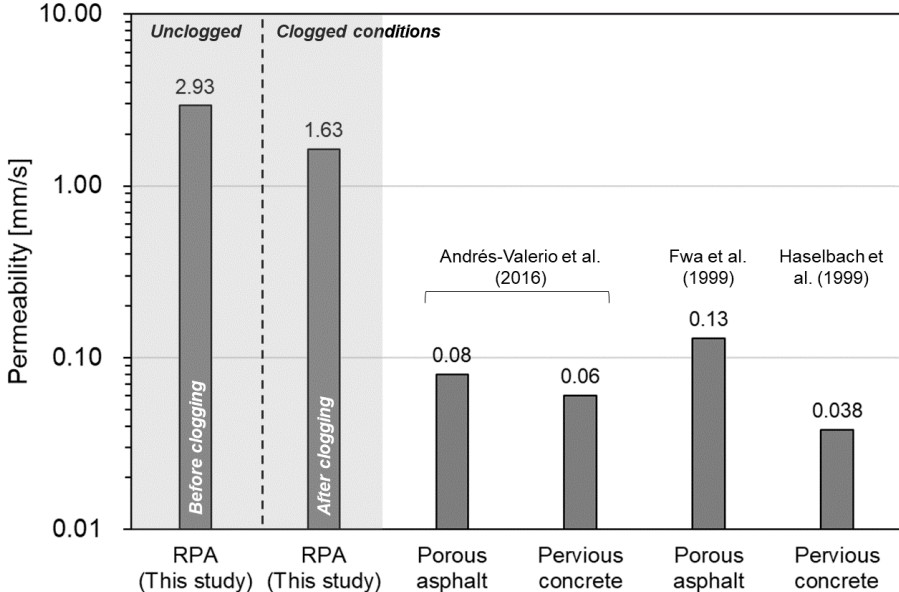

**Figure 4.** Sand-clogged permeability of the RPA block and comparison with other pervious pavement materials.

### 3.3. Durability

Figure 5 shows the average remaining mass of ten RPA block samples (square symbols) subjected to freeze–thaw durability assessment. The RPA blocks sustained <1.5% mass reduction after 60 freeze–thaw cycles, with an average loss of only 2.0% over 120 cycles (Figure 5).

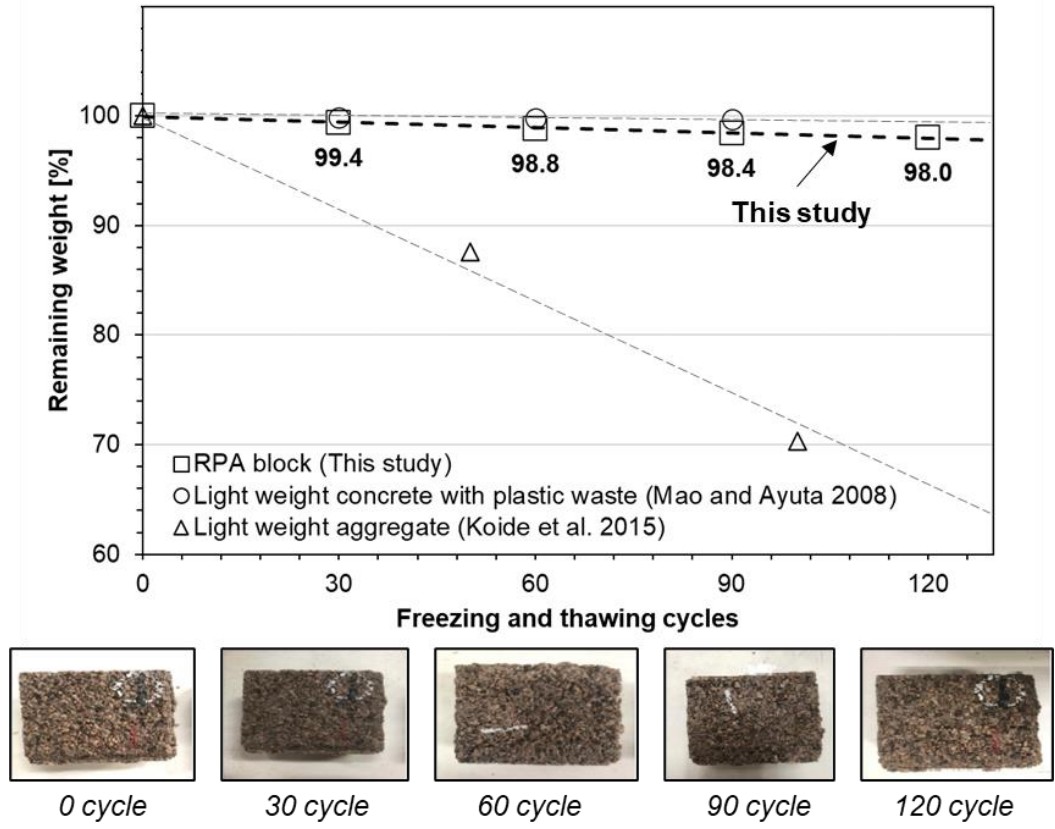

**Figure 5.** Durability (remaining mass through freeze–thaw cycles) of the RPA block and comparison with other lightweight pavement materials.

As the remaining mass of RPA blocks follows a linear reduction path, the remaining mass after 50 cycles is extrapolated to be 99.1%, which satisfies the weathering criteria (less than 1.0% per 50 cycles) for paving units (Table 1) [32]. The durability of RPA blocks is therefore adequate and at least comparable with those of lightweight aggregate and lightweight concrete involving recycled plastic-waste aggregate (Figure 5) [40,41]. Previous freeze–thaw studies of conventional and permeable pavers found that rubberized pervious concrete sustains a mass loss of 3.5% over 240 cycles, and plane concrete and clay-soil blocks sustain mass losses of 34.0% over 300 cycles and 0.7% over 28 cycles, respectively (Table 2) [42,43].

**Table 2.** Durability comparison between the RPA block and other engineered pavement materials.

| Reference | This Study | Gesoğlu et al. [42] | | Aubert and Gasc-Barbier [43] |
|---|---|---|---|---|
| Material | RPA block | Plane concrete | Rubberized pervious concrete | Clayey soil block |
| Weathered mass (%/cycles) | 2.0%/120 cycles | 34.0%/300 cycles | 3.5%/240 cycles | 0.7%/28 cycles |

### 3.4. Performance Summary and Economic Feasiblity of RPA Block

The RPA blocks satisfy international requirements for permeable pavement in terms of flexural strength, in situ permeability, and durability (Table 1), and they have higher engineering performance than other currently available permeable pavement materials. Their cost-effectiveness was estimated by comparison with porous asphalt, pervious concrete, interlocking pavers/open-cell pavers, and open-cell/grid-paving systems, assuming a unit area (1 m$^2$) implementation (Figure 6).

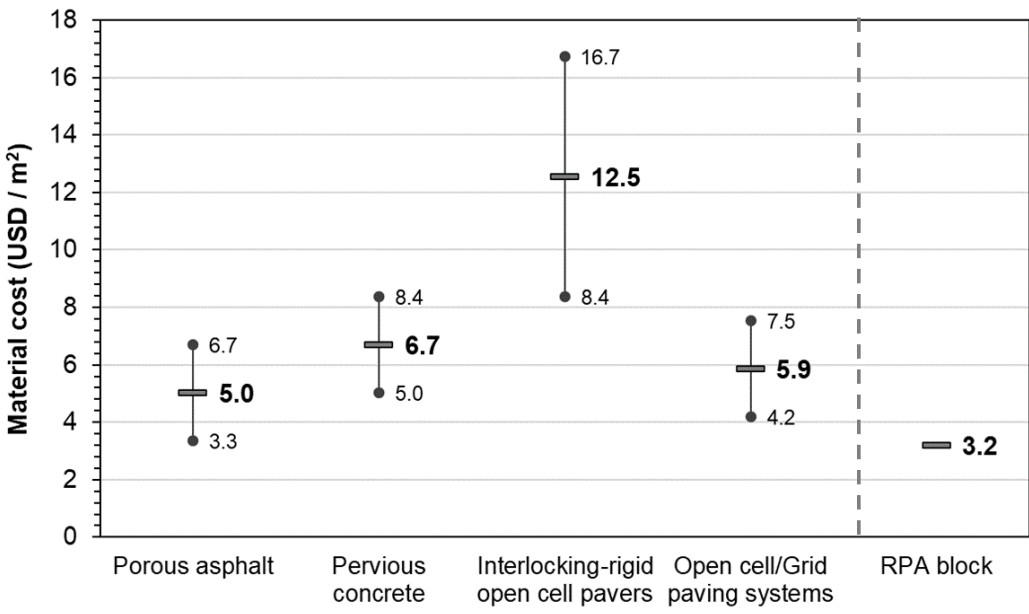

**Figure 6.** Economic feasibility (material cost) comparison between the RPA block and other permeable pavement materials. Existing permeable pavement materials are plotted with the cost range and mean value.

In Figure 6, the estimated mean material cost per 1 m$^2$ permeable pavement installation is plotted with the reported upper and lower ranges. The RPA block material cost is calculated to be USD 3.19 m$^{-2}$ by considering the material prices of PET chips, aggregates, and the fabrication process (energy for heating and machinery operation) required to produce 25 RPA blocks (200 mm × 200 mm × 80 mm). Compared to other permeable pavement materials, the RPA block seems to be economically feasible in advance. In addition, permeable pavement sites constructed with RPA blocks are expected to require lower maintenance costs due to the competitive sand-clogged permeability of RPA blocks (Table 1; Figure 3), which exceeds that of other permeable pavement systems requiring 2 to 4 times as much in situ maintenance to ensure sufficient ground infiltration [26].

Furthermore, RPA block production has the potential to reduce plastic waste by recycling PET bottles. A single block contains 81.75 g of PET chips, and their application in urban pervious paving would make a significant contribution to the reduction of global PET waste. For example, the paving of half the area of New York City (392 km$^2$) in RPA blocks would recycle 6.7% of the current global annual PET production (13 Mt) [44]. The heating of recycled PET to temperatures above 270 °C is reported to enhance its mechanical and thermal properties through crystallization [45,46], which would have contributed to the high freeze–thaw durability achieved here (Figure 5). RPA blocks are thus expected to have sufficient stability and durability even during hot seasons when pavement surface temperatures may exceed 65 °C [47,48].

## 4. Conclusions

RPA blocks made of recycled PET provide a new and environmentally friendly pervious paving material. Their flexural strength and in situ permeability are comparable with or exceed that of the most commonly used pervious-concrete paving, meeting requirements of both the USA and South Korea, with lower maintenance costs (due to less clogging) and high endurance performance in all weather conditions. Furthermore, the cost of RPA blocks is lower than that of other permeable paving systems, with reduced maintenance requirements. RPA blocks thus have high potential for sustainable use in pervious paving, with competitive engineering performance and economic feasibility.

**Author Contributions:** Conceptualization, B.-H.R. and I.C.; methodology, B.-H.R. and I.C.; validation, B.-H.R. and S.L.; formal analysis, B.-H.R., S.L., and I.C.; data curation, B.-H.R. and I.C.; writing—original draft preparation, B.-H.R.; writing—review and editing, I.C.; supervision, I.C. All authors have read and agreed to the published version of the manuscript.

**Funding:** This research was funded by the Ministry of Land, Infrastructure and Transport (MOLIT) of the Korean government through grant number 20AWMP-B114119-05 and the Korea Institute of Civil Engineering and Building Technology through the project "Development of environmental simulator and advanced construction technologies over TRL6 extreme conditions".

**Conflicts of Interest:** The authors declare no conflict of interest. The funders had no role in the design of the study; in the collection, analyses, or interpretation of data; in the writing of the manuscript, or in the decision to publish the results.

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
