# Peer review of "Pervious Pavement Blocks Made from Recycled Polyethylene Terephthalate (PET): Fabrication and Engineering Properties"

_sustainability, doi:10.3390/su12166356_

Round 1
Reviewer 1 Report
The article was written well with minimal grammar issues. The content of the manuscript was organized and easy to follow. I would recommend publication of this manuscript after responding to my comments listed below.
Lines 60-61: Please provide information about trial tests including PET/aggregate ratios and the resulted density values.
Please also provide the specific gravity of the aggregates used for this study.
Line 74: ‘along’ to ‘along with’
The unit of insitu-permeability equation in section 2.3.2 is mm/hr while the reported values in Table 1 is based on mm/sec. Please be consistent.
In Table 1, is the recommended value, the minimum or maximum value? (although this has been mentioned in section 3.2)
To be consistent and comparative, please also report the mass loss due to freeze-thaw after 50 cycles too.
Line 138: The statement shows that the mass loss after 60 cycles does not meet the recommended criteria. The authors must state the average mass loss after 50 cycles to compare it to the recommended value.
How was the mass loss calculated? Was it after the thawing cycle? In that case, was the sample dried back to its natural moisture content to compare it with the initially prepared sample? Please elaborate on this for the readership of this article.
Author Response
Authors appreciate the reviewer's time and efforts for reviewing our manuscript. All comments are addressed in the revised manuscript, accordingly. Detail point-by-point response to the reviewer's comments can be found in the attached file. Please see the attachment.

Reviewer 2 Report
The reviewed article concerns a very important problem which is plastic wastes in the global ecosystem, especially in the water environment. Thus reuse of the above-mentioned wastes is crucial from an ecological point of view. However, in my opinion, this article doesn't fit the Water Journal. It raises a problem with water pollution and presents in situ and laboratory permeability tests, but the main part of this publication is recycled PET blocks which should be used in civil engineering. The article presents research common for civil engineering. Thus, in my opinion, it should be submitted in one of the special issues of the Sustainability Journal:
https://www.mdpi.com/journal/sustainability/special_issues/Civil_Infrastructures_Sustainability_Recent
or
https://www.mdpi.com/journal/sustainability/special_issues/CivilEngineering
However, this decision should be made by the Editors. Despite this, I have reviewed this article and I have a few remarks and questions which I wrote below.
- Could the authors write something more about preliminary trials? Because in line 60 you wrote that preliminary trials indicated that 7.5% is the optimum proportion. What other proportion did you examine?
- In line 78 you wrote that the flexure test was repeated five times. How many In situ permeability, Sand-clogged permeability and Freeze-thaw durability tests you conducted? There is no such information in the article.
- Line 146, change table 2 title.
- Why in figure 5 are presented differently references than in table 2?
- In figure 5 your research results are presented by black squares. But your results contain vertical black lines ended by short horizontal lines which commonly presents statistical data. Please write what statistical data or what results are presented in figure 5 by the above-mentioned lines.
- Line 156: change number and title of Figure 3 to Figure 6
- What indicates vertical lines coming out of points in Figure 3 (6) Estimated material cost?
- How you calculated the cost-effectiveness of 1m2 ?
- Based on what date you assumed that the pervious pavement blocks made from recycled polyethylene terephthalate have the potential to utilize up to 35% of global plastic waste? Line 15.
Author Response

(The authors gave the same response as above.)

Round 2
Reviewer 1 Report
The responses are satisfactory. Therefore, I'd recommend accepting this article for publication. However. in the response to comment 2, 'g/cm3' is the unit of density not the unit weight.
Reviewer 2 Report
The authors responded to all my remarks. I don't have any other questions. The article is very interesting and should be published.